# Quantitative Multiparametric Ultrasound (mpUS) in the Assessment of Inconclusive Cervical Lymph Nodes

**DOI:** 10.3390/cancers14071597

**Published:** 2022-03-22

**Authors:** Markus H. Lerchbaumer, Katharina Margherita Wakonig, Philipp Arens, Steffen Dommerich, Thomas Fischer

**Affiliations:** 1Department of Radiology, Charité—Universitätsmedizin Berlin, Corporate Member of Freie Universität Berlin, Humboldt-Universität zu Berlin and Berlin Institute of Health, Charitéplatz 1, 10117 Berlin, Germany; markus.lerchbaumer@charite.de (M.H.L.); thom.fischer@charite.de (T.F.); 2Department of Otorhinolaryngology, Charité-Universitätsmedizin Berlin, Corporate Member of Freie Universität Berlin, Humboldt-Universität zu Berlin and Berlin Institute of Health, Campus Virchow Klinikum and Campus Charité Mitte, Charitéplatz 1, 10117 Berlin, Germany; philipp.arens@charite.de (P.A.); steffen.dommerich@charite.de (S.D.)

**Keywords:** cervical lymph nodes, multiparametric ultrasound, shear-wave elastography, head and neck squamous cell carcinoma, CEUS

## Abstract

**Simple Summary:**

Persisting cervical lymphadenopathy should be further evaluated to distinguish benign infectious enlargements from malignant diseases such as cancer of the upper airway tract, with head and neck squamous cell carcinoma or lymphatic diseases being the most common ones. Ultrasound (US) remains the primary imaging modality for assessment of cervical lymph nodes (CLN) due to their superficial localization. We aimed to investigate whether US examination and classification of CLN can be improved by adding multiparametric applications to the established B-mode–US technique by evaluating tissue stiffness and micro-vascularization. Our results show that tissue stiffness was significantly higher in malignant CLN, even in subgroups without B-mode criteria indicating malignancy. Shear wave elastography is an easy, fast and noninvasive tool available on most US devices today. This may help to detect malignant CLN with higher accuracy and help patients in cancer aftercare to detect recurrent CLN metastasis and to consecutively assess necessary treatment faster.

**Abstract:**

Background: Enlarged cervical lymph nodes (CLN) are preferably examined by ultrasound (US) by using criteria such as size and echogenicity to assess benign and suspicious CLN, which should be histologically evaluated. This study aims to assess the differentiation of malign and benign CLN by using multiparametric US applications (mpUS). Methods: 101 patients received a standardized US protocol prior to surgical intervention using B-mode–US, shear-wave elastography (SWE) and contrast-enhanced ultrasound (CEUS). SWE was assessed by 2D real-time SWE conducting a minimum of five measurements, CEUS parameters were assessed with post-processing perfusion software. Histopathological confirmation served as the gold standard. Results: B-mode–US and SWE analysis of 104 CLN (36 benign, 68 malignant) showed a significant difference between benign and malignant lesions, presenting a larger long axis and higher tissue stiffness (both *p* < 0.001). Moreover, tissue stiffness assessed by SWE was significantly higher in CLN with regular B-mode–US criteria (Solbiati Index > 2 and short-axis < 1 cm, *p* < 0.001). No perfusion parameter on CEUS showed a significant differentiation between benign and malignant CLN. Discussion: As the only multiparametric parameter, SWE showed higher tissue stiffness in malignant CLN, also in subgroups with regular B-mode criteria. This fast and easy application may be a promising noninvasive tool to US examination to ameliorate the sonographic differentiation of inconclusive CLN.

## 1. Introduction

Cervical lymphadenopathy can be caused by infections, but in case of a persistency longer than 14 days, the probability of malignancy is given, and a histological examination becomes essential [1]. Cervical lymph node (CLN) metastases are largely caused by primary malignant head and neck neoplasms of the upper aerodigestive tract [2], especially by head and neck squamous cell carcinoma (HNSCC) [3] or diseases of the lymphatic system [4]. HNSCC shows a high frequency of CLN metastasis, with cervical swellings being the primary symptom [5,6].

The appearance of CLN metastasis influences the therapeutical regime and overall prognosis [7], and CLN evaluation therefore plays an important role in cancer aftercare. In only 1% the origin of CLN, metastasis is localized extracervically (i.e., tumors of breast, lung, gastrointestinal tract, urogenital tract and central nervous system) [2]. Besides neoplastic lymphadenopathy, CLN enlargement may be caused by inflammations or viral infections and should be taken into consideration as a differential diagnosis, especially in younger patients [8].

Due to the superficial localization, B-Mode ultrasound (US) is the primary imaging modality for detection and characterization by the use of high-frequency linear transducers [9]. Well known criteria for malignant lymph node transformation are round shape and loss of fatty hilus or cystic changes and Solbiati Index (ratio long-axis/short-axis) < 2 [10,11]. Although image quality on US is increasing, the general diagnostic performance of non-enhanced US in characterization of cervical lymph nodes is still not sufficient to avoid invasive histopathological confirmation. The term “multiparametric ultrasound (mpUS)” was developed in the past years by including recent advances in US technology, yielding further parametric applications to the standard examination procedures [12]. Both shear wave elastography (SWE), which allows for metric assessment of tissue stiffness, and contrast-enhanced ultrasound (CEUS) for the depiction of microvascularization and vessel structure in real time, were frequently implemented in the diagnostic pathway for assessment of potential malignant lesions. Here, the major benefit of CEUS is real-time dynamic imaging of perfusion, and SWE is a non-invasive and fast application.

Thus, our study’s aim was to assess the value of mpUS in the differentiation of benign and malignant CLN, including SWE and parametric perfusion analysis by CEUS.

## 2. Materials and Methods

### 2.1. Patient Selection

We screened 101 patients who met the inclusion criteria: (i) 18 years or older, (ii) CLN swelling persisting more than 14 days without showing conclusive results in the serological screening for lymphotropic viruses or (iii) presentation of enlarged CLN within the oncological after care, and (iv) additional histological confirmation. Eligible patients were referred by outpatient clinic of the otorhinolaryngological department. 

Patients underwent a mpUS examination prior to histological workup, including biopsy, lymph node dissection or neck dissection. If a patient’s health did not allow for surgery, biopsies were obtained by ultrasound (US)–guided core needle biopsies.

The study was approved by the institutional ethics committee of Charité—Universitätsmedizin Berlin (protocol code EA1/087/19, date of approvement: 2 May 2019), and written informed consent was obtained from all participants according to the Declaration of Helsinki.

### 2.2. Imaging Protocol

All US examinations were performed using a standardized protocol with a multifrequency linear array transducer to assess CLN position and size. For assessment of the target lymph node, patients were examined in supine position with the head stretched upward and to the contralateral side. All examinations were performed using a single high-end US system with a 4–10 MHz multifrequency linear array transducer and a center frequency of 7 Mhz (Acuson Sequoia, Siemens Healthineers, Erlangen, Germany). For all CLN, short- and long-axis were measured, and the Solbiati Index (long-axis/short-axis) was calculated. SWE examinations were performed with optimal visualization of the target lymph node in the center of each single image. Using the 2D SWE approach, the examiner acquired five US images of each lesion with overall five consecutive SWE measurements using a circular region of interest (ROI) placed in the center of each lesion adapted to the size of the target lesion. Thus, representative tissue stiffness is given as the mean of five SWE measurements and corresponding standard deviation. ROI placement was depicted in dual-image mode for controlled assessment of the whole lesion (B-Mode image left, color-coded SWE mapping on the right, ROI for measurement visualized in both images). The standardized penetration depth was adapted to each patient for optimal visualization and correct SWE measurement. Gain was not changed to avoid potential influence on SWE measurements. CEUS examinations were performed using the same high-end systems with state-of-the-art CEUS-specific protocol and a low mechanical index (<0.1) to avoid early microbubble destruction, especially in the near field. A bolus of 1.2 mL of a second-generation ultrasound contrast agent (SonoVue^®^, Bracco Imaging, Milan, Italy) was injected intravenously followed by a 10 mL flush of 0.9%-NaCl solution. A cineloop of 90 s was stored to assess inflow and washout without movement. 

All examinations were performed by a trained otolaryngologist or an experienced radiologist. The workup is demonstrated in Figure 1.

### 2.3. Perfusion Analysis

Quantification of perfusion imaging on CEUS was performed using the post-processing software tool VueBox^®^ (Bracco Suisse SA-Software Applications, Geneva, Suisse) by using uncompressed DICOM cine loops. Both the US system and transducer were already calibrated for image analysis (predefined optimized gain adaption), and ROIs for perfusion analysis were manually placed at the borders of each lesion. ROIs did not change during the entire clip, and motion correction was used if necessary to eradicate minimal transducer movement. Intensity parameters are determined to describe the (relative) blood volume and further time-related parameter describes the mean blood flow velocity. Since there is a linear relationship between concentration of US contrast agent and intensity of the echo signal, the measurement of these parameters and corresponding AUCs allows for assessment of relative blood volume independently of microbubble inflow. The used software analysis (VueBox^®^) allow for validated quantification of relinearized video signals obtained with different based on a pixel-wise relinearization of the video signal. Besides peak parameters and AUCs, meanLin represents the median linearized signal of a region (e.g., lesion or CLN) [13]. Quantitative perfusion parameters were peak-enhancement (PE), rise time (RT), median linearized signal (meanLin), time to peak (TTP), wash-in area under the curve (WiAUC), wash-in rate (WiR), and wash-in perfusion index (WiPI). Start point of all time-related parameters was calculated by depiction of the first microbubble on the screen to avoid potential inaccurate inflow time due to different cardiac outputs. 

### 2.4. Statistical Analysis

A dependent *t* test was applied to compare after testing for normal distribution using the Kolmogorov–Smirnov test. Variables following a normal distribution are reported as mean, and associated standard deviation and subgroups were compared using the Mann–Whitney U test. Categorical variables were compared using Student’s *t* test or chi^2^ test, as appropriate. 

Mann–Whitney U tests were calculated to determine if there were differences in B-mode–US characteristics (long-axis diameter (LAD) and short-axis diameter (SAD), Solbiati Index (LAD/SAD)), SWE and CEUS parameters (as described above in the chapter “perfusion analysis”) between benign and malignant CLN. We tested for group differences in all CLN (referred to as “general cohort”), in CLN with Solbiati Index > 2 and are therefore considered to be benign following B-mode–US criteria and in small CLN with SAD < 1 cm. A two-sided significance level of α = 0.05 was defined appropriate to indicate statistical significance. Receiver operating characteristic (ROC) analyses were used to investigate the cohort’s optimal SWE cut-off values by using Youden index quantification. All statistical analyses were performed using the SPSS software (IBM Corp. Released 2016. IBM SPSS Statistics for Windows, Version 27.0. Armonk, NY, USA: IBM Corp.). 

## 3. Results

### 3.1. Study Cohort

Overall, 101 patients (35 female, 66 male) with a mean age of 60.14 years (±16.9 years) were included; of those, 99 patients had surgery (lymph node extirpation or neck dissection), and two patients underwent US-guided core needle biopsy (minimum three specimen, 14 gauge needle; Magnum™ Reusable Core Biopsy Instrument). Overall, imaging data of 104 CLN (36 benign, 68 malignant) were retrieved because we included two CLN from different sides of the neck in three patients. Thirty malign CLN were metastases of HNSCC (44.11%), 19 lymphomas (27.94%), nine malignant melanoma (13.24%) and five adenocarcinoma (7.35%), while the rest originated from other entities (see Table 1). Due to technical issues, CEUS data could only be evaluated for 98 CLN (34 benign, 64 malignant).

### 3.2. General Cohort

In the general cohort, benign CLN were smaller than malignant ones, with malignant CLN showing a longer LAD (*p* < 0.001, mean diameter of benign CLN 1.47 cm ± 0.69 cm and of malignant CLN 2.14 cm ± 1.06 cm) and a larger SAD (*p* < 0.001, mean diameter of benign CLN 0.74 cm ± 0.34 cm and of malignant CLN 1.38 cm ± 0.62 cm). The Solbiati Index in benign CLN was higher than in malignant nodes (*p* < 0.001, mean Solbiati Index of benign CLN 2.11 ± 0.81 and of malignant CLN 1.60 ± 0.46). In the general cohort, tissue stiffness implicated by SWE was higher in malignant CLN (68/104 CLN) compared to benign ones (*p* < 0.001 with a mean of 1.74 m/s ± 0.8 m/s in benign and a mean of 2.66 m/s ± 1.08 m/s in malignant CLN). The cut-off value for tissue stiffness was calculated by using ROC analyses and Youden index quantification and was determined at 2.25 m/s. Representative images are shown in Figure 1 and Figure 2, data are shown in Table 2. 

### 3.3. Subgroup Solbiati Index > 2

In the subcategorization of CLN that were classified as benign by the Solbiati Index showing a SAD/LAD ratio of >2 (*n* = 27, 17 benign and 10 malignant CLN), the SAD in malignant nodes was longer than in benign CLN (*p* = 0.031, mean SAD of 0.67 cm ± 0.34 cm in benign and 1.12 cm ± 0.62 cm in malignant CLN). No significant group differences were found for the LAD (*p* = 0.083, mean LAD of 1.76 cm ± 0.81 cm in benign and 2.63 cm ± 1.32 cm in malignant CLN). Moreover, malignant CLN (10/27) showed a higher stiffness in the group determined as benign lesions by a Solbiati Index > 2 (*p* = 0.008, mean SWE of benign CLN 1.69 m/s ± 0.57 m/s and 2.45 m/s ± 0.65 m/s in malignant CLN), and examples are depicted in Figure 3 and Figure 4. For detailed results, see Table 3.

### 3.4. Infracentimetric CLN

Considering only small CLN with a SAD < 1 cm (*n* = 49, 29 benign and 20 malignant CLN), neither the LAD (*p* = 0.968, mean in benign CLN 1.33 cm ± 0.51 cm and 1.33 cm ± 0.53 cm in malignant CLN) nor the Solbiati Index (*p* = 0.082, mean in benign CLN 2.26 ± 0.81 and 1.88 ± 0.63 in malignant CLN) showed significant group differences. For detailed results, see Table 2, Table 3 and Table 4. In the subcohort of small CLN (defined as SAD < 1 cm), malignant CLN (20/49) were stiffer than benign ones (*p* = 0.025, mean SWE of benign CLN 1.69 m/s ± 0.7 m/s and 2.27 m/s ± 0.88 m/s in malignant CLN). Variables with *p* < 0.05 are shown in Table 4.

### 3.5. CEUS

No significant group differences could be found in any of the CEUS subgroups with *p* > 0.05 for all parameters. For detailed results, see Appendix A.

### 3.6. Subgroup Analysis

All malignant entities were stiffer compared to benign nodes (mean SWE of benign CLN: 1.72 m/s ± 0.69 m/s; HNSCC: *p* < 0.001, 2.59 m/s ± 0.97 m/s; lymphoma: *p* < 0.001, 2.78 m/s ± 1.13 m/s; malignant melanoma: *p* = 0.028, 2.21 m/s ± 0.62 m/s; adenocarcinoma: *p* = 0.021, 2.54 m/s ± 0.64 m/s; other origins: *p* = 0.018, 3.15 m/s ± 1.74 m/s), whereas the analysis of B-Mode–US parameters only showed that CLN metastases originated from HNSCC (LAD: *p* < 0.001, 2.08 cm ± 0.8 cm; SAD: *p* < 0.001, 1.31 cm ± 0.58 cm; Solbiati Index: *p* < 0.001, 2.59 ± 0.97), lymphoma (LAD: *p* < 0.001, 2.41 cm ± 1.15 cm; SAD: *p* < 0.001, 1.5 cm ± 0.59 cm; Solbiati Index: *p* = 0.008, 1.6 ± 0.33) and malignant melanoma (LAD: *p* = 0.0038, 2.09 cm ± 1.02 cm; SAD: *p* < 0.001, 1.51 cm ± 0.65 cm; Solbiati Index: *p* = 0.001, 1.36 ± 0.2) and had longer LAD and SAD and a higher Solbiati Index than benign nodes (LAD 1.47 cm ± 0.69 cm, SAD 0.74 cm ± 0.34 cm, Solbiati Index 2.11 ± 0.81). CLN metastasis originating from other origins than adenocarcinoma or the ones mentioned above displayed longer SAD (*p* = 0.001, 1.74 cm ± 0.88 cm) and a higher Solbiati Index (*p* = 0.018, Solbiati Index 1.39 ± 0.36).

A significant difference of the LAD size between HNSCC and adenocarcinoma could be shown (LAD *p* = 0.0018, mean LAD in adenocarcinoma 1.39 cm ± 0.36 cm). LAD and SAD of lymphoma were bigger than in adenocarcinoma (LAD *p* = 0.012, mean LAD in lymphoma 2.41 cm ± 1.15 cm, mean LAD in adenocarcinoma 2.09 cm ± 1.02 cm; SAD *p* = 0.009, mean SAD in lymphoma 1.5 cm ± 0.59 cm, mean SAD in adenocarcinoma 1.51 cm ± 0.65 cm). CLN metastasis, which were classified as metastasis of other origins (see Table 1), were significantly bigger than adenocarcinoma (LAD: *p* = 0.045, mean LAD of adenocarcinoma 1.15 ± 0.48 cm, mean LAD of metastasis of other origins 2.84 cm ± 1.9 cm; SAD: *p* = 0.022, mean SAD in adenocarcinoma 1.51 cm ± 0.65 cm, mean SAD in metastasis of other origins 1.51 cm ± 0.65 cm). A differentiation between malignant melanoma and adenocarcinoma was also possible with LAD (*p* = 0.039, mean LAD of adenocarcinoma 1.15 ± 0.48 cm) and SAD (*p* = 0.004; mean SAD of adenocarcinoma 0.77 ± 0.17 cm).

No significant results concerning the differentiation between malignant entities of CLN could be obtained for SWE and CEUS parameters. For more detailed information, see Table 5.

## 4. Discussion

Our results implicate that B-mode–US allows for dignity estimation of CLN by assessing the LAD and SAD as well as calculating the Solbiati Index. In smaller CLN, which are referred to as CLN with SAD less than 1 cm, neither the LAD nor the Solbiati Index showed significant differences between benign and malignant CLN. The differentiation was improved by the implementation of SWE, which showed a significant group difference not only in large but also in small CLN. Although no US modality showed differences in the subgroup of malignant entities, SWE differentiated benign from any malignant CLN, showing higher tissue stiffness in malignant lesions, even in subgroups. 

Previous work showed that shear wave velocity of SWE is proportional to tissue stiffness, meaning that shear waves travel faster in stiff tissue [14], and velocity is therefore a reflection of stiffness. Metastatic lymph nodes are gaining stiffness by tumor cell infiltration [15] and tend to show an increase in hardness and density as well as vascularization before changing in size [16]. This could explain why in our cohort SWE was superior to the Solbiati Index, especially in the group of CLN with a SAD less than 1 cm. Our results support previous findings stating that higher velocity (and therefore higher stiffness) is a sign of malignancy [17,18], with the mean SWE in our cohort in benign CLN reported as 1.72 m/2 (±0.69 m/s) and the mean SWE of malignant CLN with 2.63 m/s (±1.03 m/s). 

As regional lymph node metastases have a high influence on treatment and prognosis, a decision on the treatment of the neck always needs to be considered. In HNSCC, there is a high risk of occult CLN metastasis in clinically nodal negative necks with incidences of 30% in oropharyngeal [19], 23.7% in hypopharyngeal [20] and 20.5% in laryngeal squamous cell carcinoma (SCC) [21]. It is not always an easy decision in patients with a clinically nodal negative neck between overtreatment in terms of a complete or even a contralateral neck dissection with risks such as bleeding or intraoperative nerve damage and missing occult metastasis [22]. CLN with a SAD less than 1 cm are primarily rated as non-suspicious [23]. In our cohort, out of 49 CLN with a SAD smaller than 1 cm, 29 CLN were benign and 20 malignant (40%), which indicate a high risk of overseeing occult metastasis. Our analysis showed that in this specific CLN group, no significant differences between malignant and benign CLN were found using the Solbiati Index, but they may be identified by SWE. This is in line with findings of Guan and colleagues, who demonstrated SWE as a proper modality to detect small CLN metastases in patients with HNSCC of the nasopharynx [24]. In 2016, Pauzie et al. showed that not only large CLN but also infracentimetric CLN metastases in HNSCC are correlated with a poor prognosis concerning disease-free survival and suggest close postoperative surveillance of the neck [25].

This underlines the importance of close patient monitoring and regular assessment of CLN as part of oncological aftercare to detect recurrent metastasis as quick as possible [26]. 

CEUS analysis did not show any significant results, although one would expect clear visualization differences in the vascularization patterns of benign and malignant CLN, with one of the most common US–B-Mode criterion for malignancy being the absence of the hilus sign [27]. An explanation for the absence of conclusive CEUS results in our cohort could be the large amount of metastatic CLN with partial necrosis, which understandably has an influence on the quantitative determination of the vascularization patterns and parametric assessment of whole-lesion perfusion. Although metastatic nodes may differ optically from benign CLN, the software only registers the amount of contrast-enhanced bubbles and the perfusion pattern appearing in the region of interest. This imbalance of necrosis and different inflow characteristics influences the post-processing analysis. Despite areas of necrosis in larger CLN, which would be visible without post-processing perfusion analysis, especially since smaller CLN potentially have full contrast enhancement without necrotic areas. Nevertheless, dynamic perfusion patterns (besides the amount of lesion contrast enhancement) such as wash-in or wash-out curves were not able to differentiate benign from malignant CLN in our cohort. While we could not show any CEUS differences, other study groups retrieved good results with this technique, especially in the differentiation of thyroid cancer metastasis from reactive CLN [28].

US is an established modality to assess CLN, and both B-mode–US and SWE are useful in distinguishing between malignant and benign CLN, whereas the classical use of the Solbiati Index seems to be limited in the differentiation of infracentimetric CLN. In this field, the measurement of tissue stiffness on SWE may be a promising additional parameter. SWE is a noninvasive and quick addition to the US examination and was superior in differentiating benign from malignant CLN compared to non-enhanced B-mode characteristics and was even useful in differentiation of small CLN. It can easily be added to the regular US examination, not only before surgery, but also as part of oncological after care.

## 5. Conclusions

SWE showed a superior differentiation of inconclusive CLN not only compared to CEUS perfusion parameters but also to US–B-mode criteria. Thus, non-invasive SWE assessment could be implemented in each regular US examination. Further research is needed to confirm these results and to implement SWE as a new parameter in the characterization of lymph nodes in general. 

## Figures and Tables

**Figure 1 cancers-14-01597-f001:**
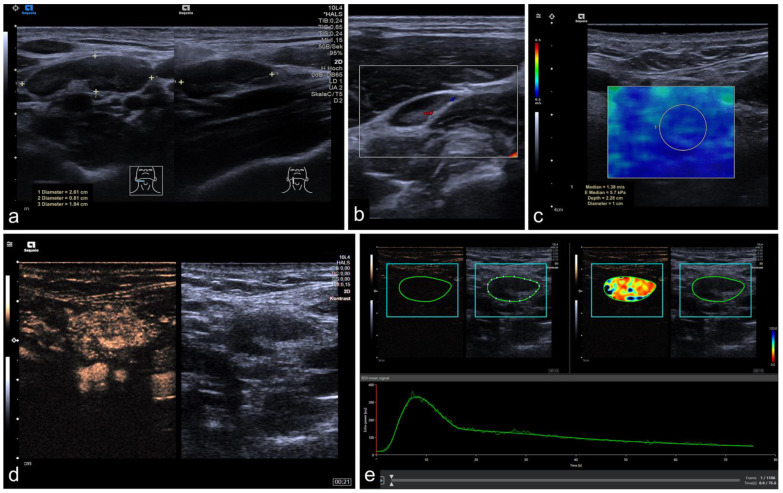
Representative image of mpUS workflow. (**a**) Long-axis and short-axis diameter measured in dual mode, (**b**) color-coded Doppler US depicting macrovascularity, (**c**) SWE measured by circular ROI revealed low tissue stiffness (1.38 m/s), (**d**) CEUS image 21 s after contrast injection with regular enhancement, and (**e**) RAW data perfusion analysis with demarcation margins (blue) and lesion ROI (green) showing the corresponding time-intensity curve and color-coded map. The CLN was confirmed as benign.

**Figure 2 cancers-14-01597-f002:**
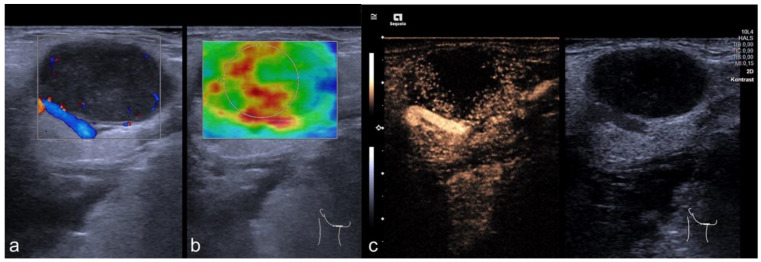
Representative image of mpUS. CLN suspicious for cervical manifestation of a lymphoma showed round shaped; hypoechogenic lymph nodes (**a**) with high tissue stiffness on SWE (**b**) and irregular contrast enhancement showed a large area of central necrosis (**c**). Final diagnosis was HNSCC, indicated by high stiffness on SWE.

**Figure 3 cancers-14-01597-f003:**
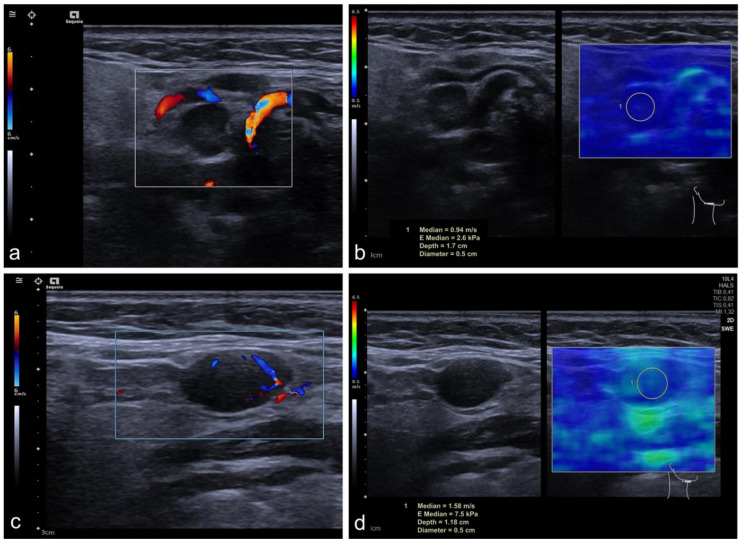
Inconclusive CLN on US and SWE. The figure demonstrates two cases of inconclusive CLN. Both CLN nodes were round shaped and represented decreased long-axis diameter combined with a Solbiati index < 2, thus indicating malignancy or inconclusive (Case 1: (**a**,**b**); Case 2: (**c**,**d**)). The abnormal vascularization of color-coded duplex sonography is demonstrated as a non-parametric imaging marker (Figure 2a,c). Both CLN presented low stiffness on SWE as demonstrated in Figure 2b for Case 1 (0.94 m/s) and Case 2 in Figure 2d (1.48 m/s), despite their abnormalities on non-enhanced US (**b**,**d**).

**Figure 4 cancers-14-01597-f004:**
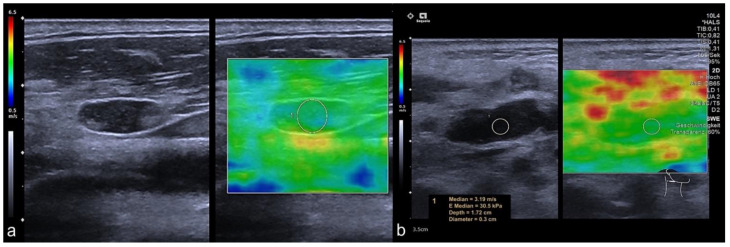
Representative SWE approach in regular CLN on B-mode US. Both CLN showed regular Solbiati Index > 2 without decreased long-axis diameter. Shear wave velocity on SWE was slightly high in Case 1 (**a**) and high in Case 2 (**b**) visualized by color-coded 2D-SWE maps and clarified by a value of 3.19 m/s in Case 2 (Figure 2b). Both CLN showed no parametric abnormalities in B-Mode US. Nevertheless, tissue stiffness on SWE was significantly increased compared to benign CLN, as representatively shown in Figure 2.

**Table 1 cancers-14-01597-t001:** Study Population. Abbreviations: HNSCC, head and neck squamous cell carcinoma; CLL, chronic lymphocytic leukemia; SWE, shear wave elastography; CEUS, contrast enhanced ultrasound, CLN, cervical lymph nodes. Continuous variables are given as mean (SD), categorical variables are given as absolute/total numbers (n/N), and percentages in brackets.

**Patients**	**101**
Female	35/101
Male	66/101
Mean age	60.14 years (±16.9 years)
**CLN**	**104**
benign	36/104
malign	68/104
HNSCC	30/104
Lymphoma	19/104
Malignant melanoma	9/104
Adenocarcinoma (breast, salivary gland, lung)	5/104
Prostate cancer	1/104
Transitional cell carcinoma	1/104
Atypical fibroxanthoma	1/104
CLL	1/104
Renal cell cancer	1/104
**SWE**	**104 CLN**
Benign	36/104
malignant	68/104
**CEUS**	**98 CLN**
Benign	34/98
malignant	64/98

**Table 2 cancers-14-01597-t002:** Analysis of general cohort. All metric variables are given as mean with corresponding standard deviation. Abbreviations: SWE, shear wave elastography.

Metric Parameters	General Cohort		
	Benign (*n* = 36)	Malignant (*n* = 68)	*p*-value (*n* = 104)
**Long Axis Diameter (cm)**	1.47 ± 0.69	2.14 ± 1.06	<0.001
**Short Axis Diameter (cm)**	0.74 ± 0.34	1.38 ± 0.62	<0.001
**Solbiati Index** *	2.11 ± 0.81	1.60 ± 0.46	<0.001
**Shear Wave Elastography (*n* = 104)**			
**SWE (m/s)**	1.72 ± 0.69	2.63 ± 1.03	<0.001

* defined as ratio of long axis to short axis.

**Table 3 cancers-14-01597-t003:** Analysis of subgroup “Solbiati Index > 2”. All metric variables are given as mean with corresponding standard deviation. Abbreviations: SWE, shear wave elastography.

Metric Parameters	Solbiati Index > 2		
	Benign (*n* = 17)	Malignant (*n* = 10)	*p*-value (*n* = 27)
**Long Axis Diameter (cm)**	1.76 ± 0.81	2.63 ± 1.32	> 0.05
**Short Axis Diameter (cm)**	0.67± 0.34	1.12 ± 0.62	0.031
**Solbiati Index** *	-	-	-
**Shear Wave Elastography (*n* = 27)**			
**SWE (m/s)**	1.69 ± 0.57	2.45 ± 0.65	0.008

* defined as ratio of long axis to short-axis.

**Table 4 cancers-14-01597-t004:** Analysis of subgroup “Short axis diameter < 1 cm”. All metric variables are given as mean with corresponding standard deviation. Abbreviations: SWE, shear wave elastography.

Metric Parameters	Short Axis Diameter < 1 cm		
	Benign (*n* = 29)	Malignant (*n* = 20)	*p*-value (*n* = 49)
**Long Axis Diameter (cm)**	1.33 ± 0.51	1.33 ± 0.53	> 0.05
**Short Axis Diameter (cm)**	-	-	-
**Solbiati Index** *	2.26 ± 0.81	1.88 ± 0.63	> 0.05
**Shear Wave Elastography (*n* = 49)**			
**SWE (m/s)**	1.69 ± 0.7	2.27 ± 0.88	0.025

* defined as ratio of long axis to short axis.

**Table 5 cancers-14-01597-t005:** All metric variables are given as median with corresponding standard deviation. Only values with a *p* value < 0.05 are shown. The group that is compared to the subgroups is printed in bold letters. Abbreviations: SWE, shear wave elastography; HNSCC, head and neck squamous cell carcinoma; CLN, cervical lymph node; CEUS, contrast enhanced ultrasound; LAD, long-axis diameter; SAD, short-axis diameter; *n*, number of lymph nodes included.

Tumor Entity	LAD (cm)	SAD (cm)	Solbiati Index *	SWE (m/s)	CEUS
Benign CLN (*n* = 36)	1.47 ± 0.69	0.74 ± 0.34	2.11 ± 0.81	1.72 ± 0.69	
HNSCC (*n* = 30)	*p* < 0.001 2.08 ± 0.8	*p* < 0.001 1.31 ± 0.58	*p* = 0.023 1.73 ± 0.54	*p* < 0.001 2.59 ± 0.97	*p* > 0.05
Lymphoma (*n* = 19)	*p* < 0.001 2.41 ± 1.15	*p* < 0.001 1.5 ± 0.59	*p* = 0.008 1.6 ±0.33	*p* < 0.001 2.78 ± 1.13	*p* > 0.05
Malignant melanoma (*n* = 9)	*p* = 0.038 2.09 ± 1.02	*p* < 0.001 1.51 ± 0.65	*p* = 0.001 1.36 ± 0.2	*p* = 0.028 2.21 ± 0.62	*p* > 0.05
Adenocarcinoma (*n* = 5)	*p* = 0.36 1.15 ± 0.48	*p* = 0.265 0.77 ± 0.17	*p* = 0.069 1.48 ± 0.59	*p* = 0.021 2.54 ± 0.64	*p* > 0.05
CLN metastasis of other origins (*n* = 5)	*p* = 0.15 2.56 ± 1.99	*p* = 0.001 1.74 ± 0.88	*p* = 0.018 1.39 ± 0.36	*p* = 0.018 3.15 ± 1.74	*p* > 0.05
**HNSCC**					
Adenocarcinoma	*p* = 0.018	*p* = 0.059	*p* = 0.109	*p* = 0.962	*p* > 0.05
**Lymphoma**					
Adenocarcinoma	*p* = 0.012	*p* = 0.009	*p* = 0.145	*p* = 0.915	*p* > 0.05
CLN metastasis of other origins	*p* = 0.619	*p* = 0.546	*p* = 0.189	*p* = 0.972	*p* > 0.05
**Malignant Melanoma**					
Adenocarcinoma	*p* = 0.039	*p* = 0.004	*p* = 0.549	*p* = 0.423	*p* > 0.05
CLN metastasis of other origins	*p* = 0.947	*p* = 0.593	*p* = 0.641	*p* = 0.257	*p* > 0.05
**Adenocarcinoma**					
CLN metastasis of other origins	*p* = 0.076	*p* = 0.036	*p* = 0.754	*p* = 0.754	*p* > 0.05

* defined as ratio of long axis to short axis.

## Data Availability

The datasets generated and analyzed during the current study are not publicly available but are available from the corresponding author on reasonable request.

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
