# Peer review of "Quantitative Multiparametric Ultrasound (mpUS) in the Assessment of Inconclusive Cervical Lymph Nodes"

_cancers, 2022, doi:10.3390/cancers14071597_

Round 1

Reviewer 1 Report

The authors present an interersting paper showing the power to discern healthy form pathologic cervical lymph nodes.

Assessing a patient cohort in a clinical study shows that state-of-the-art ultrasound imagery using shear-wave-elastography, has a statistically significant power to detect malignancies.

The manuscript is generally fine, some adaptatioins, however, are required.

So, in all Tables the entity "Lin" needs to be specified. Moreover, all numbers need do carry the appropriate SI-unit to allow understanding.

In all statistical evaluations when an error of the first kind of 5% is set, only values smaller than this one need to be given in order to ease readability of the tables.

Concerning the Materials and Methods section, it is recommended to show at each step of the workflow appropriate images as set on the US device so that the reader may see the respective acquired US image.

The authors might consider merging the tables 2a and 2b, and 2c into one to allow an easy apprehension of the results.

The legend to Fig. 2 needs improvement, the content might eventually be adapted in such a way to show similar information as Fig. 1. Similarily to Fig. 3.

Table 3 will gain understandability and make the results of the work easily perceptible if only p-valuse smaller than 0.05 are given; more specifically, it will reveal that only SAD is a biomarker for adenocarcinoma only, SWE for all pathologies, and the Solbitin Index for all entities but adenocarcinoma.

Author Response

The authors present an interersting paper showing the power to discern healthy form pathologic cervical lymph nodes. Assessing a patient cohort in a clinical study shows that state-of-the-art ultrasound imagery using shear-wave-elastography, has a statistically significant power to detect malignancies. The manuscript is generally fine, some adaptatioins, however, are required.

  1. So, in all Tables the entity "Lin" needs to be specified. Moreover, all numbers need do carry the appropriate SI-unit to allow understanding.

A: Thank you for this advice. We clarified the term “Lin” and explained it more precisely. In General, CEUS parameters do not follow a regular SI-Unit. The linearized CEUS signal is based on intensity and time parameters on post-processing.

In all statistical evaluations when an error of the first kind of 5% is set, only values smaller than this one need to be given in order to ease readability of the tables.

A: Thank you for this comment. We changed the tables as recommended to show only parameters with significant p-Values <0.05. Thus may further improve readability.

Concerning the Materials and Methods section, it is recommended to show at each step of the workflow appropriate images as set on the US device so that the reader may see the respective acquired US image.

A: Thank you for this comment. We agree with the reviewer that this step would further improve the understanding and teaching point of this paper. We introduced a figure (Figure 2) to improve understanding of mpUS and our structured protocol.

“Line 116: Figure 1. Representative image of mpUS workflow. a) Long-axis and short-axis diameter measured in Dual mode, b) color-coded Doppler US depicting macrovascularity, c) SWE measured by circular ROI revealed low tissue stiffness (1.38 m/s), d) CEUS image 21 sec-onds after contrast injection with regular enhancement, e) RAW data perfusion analysis with demarcation margins (blue) and lesion ROI (green) showing the corresponding time-intensity curve and color-coded map. The CLN was confirmed as benign.”

The authors might consider merging the tables 2a and 2b, and 2c into one to allow an easy apprehension of the results.

A: Thank you for this comment. We kindly agree, that splitting these high-volume tables may improve the readability. Therefore we structured the tables as recommended into Table 2, 3 and 4.

The legend to Fig. 2 needs improvement, the content might eventually be adapted in such a way to show similar information as Fig. 1. Similarily to Fig. 3.

A: Thank you for this important comment. We improved both figure legends as recommended.

Line 218: “Fig. 2: Figure 2 . Inconclusive cLN on US and SWE. The figure demonstrate two cases of inconclu-sive cLN. Both cLN nodes were round shaped and represented decreased long-axis diameter combined with a Solbiati-index <2 – thus indicating malignancy or inconclusive (Case 1: a, b; Case 2: b,d). The abnormal vascularization of color-coded duplex sonography is demonstrated as a non-parametric imaging marker (Fig. 2a and 2c). Both CLN presented low stiffness on SWE as demonstrated in Figure 2b for Case 1 (0.94 m/s) and Case 2 in Figure 2d (1.48 m/s) despite their abnormalities on non-enhanced US (b,d).”

“Fig. 3: Figure 3.  Representative SWE approach in regular cLN on B-mode US. Both cLN showed regular Solbiati-Index > 2 without decreased long-axis diameter. Shear wave velocity on SWE was slighty high in Case 1 (a) and high in Case 2 (b) visualized by color-coded 2D-SWE maps and clarified by a value of 3.19 m/s in Case 2 (Figure 2b). Both cLN showed no parametric abnormalities in B-Mode US. Nevertheless, tissue stiffness on SWE was significantly increased compared to be-nign cLN as representative shown in Figure 1.”

Table 3 will gain understandability and make the results of the work easily perceptible if only p-valuse smaller than 0.05 are given; more specifically, it will reveal that only SAD is a biomarker for adenocarcinoma only, SWE for all pathologies, and the Solbitin Index for all entities but adenocarcinoma.

A: Thank you for this important comment. We changed Table 3 as recommended.

Reviewer 2 Report

This interesting study evaluated the value of multiparametric ultrasound (mpUS) in the assessment of 104 cervical lymph nodes from 101 patients and showed that tissue stiffness was significantly higher in malignant cervical lymph nodes.

Major Remarks

Material and Methods:

P4, L148: “US-guided biopsy” Did you use core needle biopsy or fine needle aspiration? Please specify.

P4, L163: “In the subcategorization of CLN that were classified as benign by the Solbiati-Index…” Please make a separate paragraph for this group. This subcategory is the most relevant one, because here a definitive answer regarding to malignancy is often impossible.

P2, L62: In the introduction it was cited that “Well known criterias for malignant lymph node transformation are round shape 62 and loss of fatty hilus or cystic changes and Solbiati-Index (ratio long-axis/short-axis) < 2 63 [10,11].” In fact, you did not evaluate the criteria “loss of fatty hilus” or “cystic changes”. Please provide this additional data.

Results:

P8, L219: “3.1. CEUS, followed by 3.4 Subgroup Analysis.” Please check numbers of paragraphs.

Discussion

Hypoechogenicity and necrotic areas are highly important morphological features in malign lymph nodes. It would be very interesting to know the added value of SWE in lymph nodes without obvious “hpoechogenicity”, “loss of fatty hilus” or “cystic changes”. Please correlate these findings with SWE, and most important, the absence of all malign features with SWE.  

Please provide a cut-off value for tissue stiffness indicating malignancy.

Author Response

This interesting study evaluated the value of multiparametric ultrasound (mpUS) in the assessment of 104 cervical lymph nodes from 101 patients and showed that tissue stiffness was significantly higher in malignant cervical lymph nodes.

Material and Methods:

  1. P4, L148: “US-guided biopsy” Did you use core needle biopsy or fine needle aspiration? Please specify.

A: Thank you for this advice. We defined this topic in detail to specify the form of biopsy.

Line 163: Overall 101 patients (35 female, 66 male) with a mean age of 60.14 years (± 16.9 years) were included, of those, 99 patients had surgery (lymph node extirpation or neck dissection) and two patients underwent US-guided core needle biopsy (Minimum three specimen, 14 gauge needle; Magnum™ Reusable Core Biopsy Instrument).

  1. P4, L163: “In the subcategorization of CLN that were classified as benign by the Solbiati-Index…” Please make a separate paragraph for this group. This subcategory is the most relevant one, because here a definitive answer regarding to malignancy is often impossible.

A: Thank you for this important advice. We agree with the reviewer that this subgroup is of high interest – therefore we implemented a new paragraph as recommended. 

Line 189: 3.3. Subgroup Solbiati-Index > 2

“In the subcategorization of CLN that were classified as benign by the Solbiati-Index showing a SAD/LAD ratio of > 2 (n = 27, 17 benign and 10 malignant CLN) the SAD in malignant nodes was longer than in benign CLN (p = 0.031, mean SAD of 0.67 cm ± 0.34 cm in benign and 1.12 cm ± 0.62 cm in malignant CLN). No significant group differences were found for the LAD (p = 0.083, mean LAD of 1.76 cm ± 0.81 cm in benign and 2.63 cm ± 1.32 cm in malignant CLN). Moreover, malignant CLN (10/27) showed a higher stiffness in the group determined as benign lesions by a Solbiati-Index > 2 (p = 0.008, mean SWE of benign CLN 1.69 m/s ± 0.57 m/s and 2.45 m/s ± 0.65 m/s in malignant CLN), examples are depicted in Figure 3 and 4. For detailed results see Tables 3.“

Line 203: 3.4 Infracentrimetric CLN

“Considering only small CLN with a SAD < 1cm (n = 49, 29 benign and 20 malignant CLN) neither the LAD (p = 0.968, mean in benign CLN 1.33 cm ± 0.51 cm and 1.33 cm ± 0.53 cm in malignant CLN) nor the Solbiati-Index (p = 0.082, mean in benign CLN 2.26 ± 0.81 and 1.88 ± 0.63 in malignant CLN) showed significant group differences. For detailed results see Tables 2a, b and c. In the subcohort of small CLN (defined as SAD < 1 cm), ma-lignant CLN (20/49) were stiffer than benign ones (p = 0.025, mean SWE of benign CLN 1.69 m/s ± 0.7 m/s and 2.27 m/s ± 0.88 m/s in malignant CLN). Variables with p<0.05 are shown in Table 4.”

  1. P2, L62: In the introduction it was cited that “Well known criterias for malignant lymph node transformation are round shape 62 and loss of fatty hilus or cystic changes and Solbiati-Index (ratio long-axis/short-axis) < 2 63 [10,11].” In fact, you did not evaluate the criteria “loss of fatty hilus” or “cystic changes”. Please provide this additional data.

A: Thank you for this important topic. Loss of fatty hilus is known as a qualitative, non-parametric variable. Since this work is focused on parametric imaging including B-Mode US, Doppler Imaging, SWE and CEUS – we decided to implement only (para)metric data. Since US is known as an operator dependant modality, the submitted work shout clarify how metric data without subjectivity may improve the diagnostic algorithm and performance in the assessment of cLN. Thus, we decided to implement “fatty hilus loss” in another publication focussing on qualitative assessment with mpUS by two experienced radiologists. The amount of data (if including metric and qualitative parameters) would be too much and may be not suitable for one submission.

 Results:

P8, L219: “3.1. CEUS, followed by 3.4 Subgroup Analysis.” Please check numbers of paragraphs.

 A: Thank you for this comment. We changed as recommended to ensure the correct order of subchapters.

Discussion

Hypoechogenicity and necrotic areas are highly important morphological features in malign lymph nodes. It would be very interesting to know the added value of SWE in lymph nodes without obvious “hpoechogenicity”, “loss of fatty hilus” or “cystic changes”. Please correlate these findings with SWE, and most important, the absence of all malign features with SWE.  

A: Thank you for this advice. We tried to meet this concern by evaluating both CEUS and SWE in CLN smaller 1cm as well as Solbiati-Index >2. All mentioned lymph-nodes included in these subgroups are without any changes indicating malignancy. Areas of necrosis are only clearly visualized after contrast-injection. In this submission, we focussed on metric data only and avoided moreover subjective findings (We will work on this subproject in another publication). Furthermore, including all non-parametric findings would increase the length of this manuscript avoiding a focus on parametric imaging. 

Please provide a cut-off value for tissue stiffness indicating malignancy.

A: Thank you for this advice. We added the recommended parameters in our Results section.

Round 2

Reviewer 2 Report

The authors sufficiently addressed to my queries and improved the manuscript.